# Antiproliferative Benzoindazolequinones as Potential Cyclooxygenase-2 Inhibitors

**DOI:** 10.3390/molecules24122261

**Published:** 2019-06-18

**Authors:** Aurora Molinari, Alfonso Oliva, Marlene Arismendi-Macuer, Leda Guzmán, Waldo Acevedo, Daniel Aguayo, Raúl Vinet, Arturo San Feliciano

**Affiliations:** 1Instituto de Química, Facultad de Ciencias, Pontificia Universidad Católica de Valparaíso, Valparaíso 2373223, Chile; alfonso.oliva@pucv.cl (A.O.); m.arismendi.m@gmail.com (M.A.-M.); leda.guzman@pucv.cl (L.G.); waldo.acevedo@pucv.cl (W.A.); 2Centro de Bioinformática y Biología Integrativa, Facultad de Ciencias de la Vida, Universidad Nacional Andrés Bello, Santiago 8370146, Chile; daniel.aguayo@unab.cl; 3Laboratorio de Farmacología, Centro de Micro Bioinnovación, Facultad de Farmacia, Universidad de Valparaíso, Valparaíso 2360102, Chile; raul.vinet@uv.cl; 4Centro Regional de Estudios en Alimentos Saludables (CREAS), Valparaíso 2362696, Chile; 5Departamento de Ciencias Farmacéuticas-Química Farmacéutica, Facultad de Farmacia, CIETUS, IBSAL, Universidad de Salamanca, 37007 Salamanca, Spain; artsf@usal.es

**Keywords:** 1*H*-benzo[*f*]indazole-4,9-diones, benzoindazolequinones, antiproliferative activity, COX-2 inhibitors, docking

## Abstract

Quinones and nitrogen heterocyclic moieties have been recognized as important pharmacophores in the development of antitumor agents. This study aimed to establish whether there was any correlation between the in silico predicted parameters and the in vitro antiproliferative activity of a family of benzoindazolequinones (BIZQs), and to evaluate overexpressed proteins in human cancer cells as potential biomolecular targets of these compounds. For this purpose, this study was carried out using KATO-III and MCF-7 cell lines as in vitro models. Docking results showed that these BIZQs present better binding energies (ΔG*_bin_*) values for cyclooxygenase-2 (COX-2) than for other cancer-related proteins. The predicted ∆G*_bin_* values of these BIZQs, classified in three series, positively correlated with IC_50_ measured in both cell lines (KATO-III: 0.72, 0.41, and 0.90; MCF-7: 0.79, 0.55, and 0.87 for Series I, II, and III, respectively). The results also indicated that compounds **2a**, **2c**, **6g**, and **6k** are the most prominent BIZQs, because they showed better IC_50_ and ∆G*_bin_* values than the other derivatives. In silico drug absorption, distribution, metabolism, and excretion (ADME) properties of the three series were also analyzed and showed that several BIZQs could be selected as potential candidates for cancer pre-clinical assays.

## 1. Introduction

Cancer is the second leading cause of mortality worldwide, accounting for 9.6 million deaths in 2018. Globally, nearly one in nine deaths is due to cancer, where breast cancer (BC) and gastric cancer (GC) are among the cancers with the highest mortality rates in the world [1]. Cancer development is a complex process that usually takes many years to progress through various stages before its clinical presentation [1,2]. Proper cancer treatment usually requires combined and aggressive therapeutic strategies including surgery, radiotherapy, biological therapy, chemotherapy, and combinations of them [2]. Chemotherapy has been the backbone in cancer treatments, but unfortunately, multidrug resistance is the major factor in the failure of many forms of chemotherapy [3]. Several molecular mechanisms promote or enable drug resistance, such as drug inactivation, drug target alteration, drug efflux, DNA damage repair and mutation in target gene leading to cell death inhibition. In addition, inherent tumor cell heterogeneity plays a role in drug resistance [4]. Therefore, it is urgent to find new active molecules to evaluate their therapeutic effectiveness [5].

Natural and synthetic quinones have significant biological activities, including antitumor properties, which explain their clinical use as drugs to treat cancer [6]. Among them, doxorubicin is one of the most used anticancer chemotherapeutic agents to treat solid tumors and acute leukemia, however, its use has been correlated with a high risk of cardiomyopathy [7].

Indazole derivatives also represent a large family of interesting cancer-related drugs because indazole moieties are present in molecules with a wide variety of biological activities [8]. These molecules have, in their structures, aromatic heterocycles difficult to find in natural products and therefore the chemical synthesis is the main route to obtain them [9]. Among these families, 1*H*-Benzo[*f*]indazole-4,9-quinones (BIZQs) are obtained from the 1,3-dipolar addition of diazomethanes to 1,4-naphthoquinones and have been used as scaffolds to design new anticancer molecules [10]. These compounds can also be prepared by direct condensation-cyclization reaction of 2-acetyl-6-(4-methyl-3-pentenyl)-1,4-naphthoquinone with hydrazines, as it has been previously described by Molinari et al. [11]. The indazolediones have a cytotoxic effect against several types of cancer cells, such as L1220 murine leukemia, MCF-7 breast carcinoma, PC-3 prostate carcinoma, and MKN-45 gastric adenocarcinoma [12]. Due to the coplanar polycyclic structure of these molecules, their cytostatic action is associated with the ability to intercalate in the DNA strands and to generate reactive oxygen species (ROS), which can damage biomolecules and inhibit mitochondrial function.

Another critical consideration when synthesizing new anticancer drugs is to facilitate their transport across the cell membrane, which could be achieved by conjugation with amino acids [13]. Regarding this subject, we have reported on the synthesis of a series of twenty-four new BIZQs, including several conjugated with Gly, Ala, Phe, and Glu (Figure 1), and most of these derivatives showed antiproliferative activity on two types of human cancer cells as KATO-III gastric carcinoma (GC) and MCF-7 breast carcinoma (BC) [14].

To complement the research, this study aimed to attain insight on the mechanism of action of the BIZQs and to correlate the in vitro experimental results with those from several in silico studies, mainly, those of molecular docking with proteins involved in carcinogenic process, such as cyclooxygenase-2 (COX-2), mitogen-activated protein kinase (MAPK-1), tyrosine protein kinase (TPK-JAK), vascular endothelial growth factor receptor 2 (VEGFR-2), and estrogen receptors (ERs) among others, to identify possible target proteins for these BIZQs [15,16,17,18,19,20,21,22].

It has been reported that COX-2 overexpression is correlated with most inflammatory processes and particularly with chronic inflammation-related cancers and metastasis in GC and BC [23,24,25]. For instance, GC has been linked to chronic inflammation due to *Helicobacter pylori* infection and colorectal cancer to chronic bowel inflammatory disease [15,26,27]. Predictions of some relevant physicochemical parameters, intrinsic bioactivity, drug-likeness, toxicity properties, and ADME descriptors for the BIZQs are also analyzed.

## 2. Results and Discussion

### 2.1. Chemistry

The twenty-four 1*H*-benzo[*f*]indazole-4,9-dione derivatives, **2a** to **6m**, were classified into Series I, II and III according to their substitution patterns at position N^1^ of the indazole fragment, and into ***subseries a*** and ***b*** according to the absence (BIZQs **2a**–**5c**) or presence (BIZQs **6a**–**m**) of one conjugated amino acid residue in the side-chain attached to position C-7, respectively (Figure 1). Compounds of Series I have no substituent at N^1^, while those of Series II have a 2-hydroxyethyl group and those of Series III contain a 2-acetoxyethyl group. In compounds **2a**–**c** to **5a**–**c** (***subseries a***), the substituent R_1_ contains a 2-methylpropenyl radical or its epoxy derivative, or its degraded aldehyde or carboxylic groups. Compounds **6a**–**m** (***subseries b***), are benzoindazolequinones conjugated with some C-protected amino acids as Gly, Ala, Phe, and Glu [11,14] (Figure 1).

It is expected that BIZQs conjugation with different amino acids allows to increase the affinity with the l-type amino acid transporter 1 (LAT1), overexpressed on the membrane of various tumor cells, facilitating the transport of molecules through the cell membrane, according to the strategy explored by Wu et al. [13].

### 2.2. In Silico Virtual Screening for Potential Antineoplastic Targets of BIZQs

The physicochemical and pharmacological characteristics and other properties of the BIZQs were assessed using several in silico methods.

First, aiming to reach some insight on the intrinsic bioactivity of BIQZs, their structures were subjected to analysis using Molinspiration algorithms [28], obtaining interesting results on their bioactivity prediction. Globally, all the BIZQs would be active as enzyme inhibitors, with scores ranging between 0.36 (**5c**) and 0.63 (**3a**), for the simpler BIZQs **2a**–**5c**, and between 0.15 (**6i**, **6k**) and 0.30 (**6d**) for those conjugated BIZQs **6a**–**6m**. Besides, several compounds, mainly those belonging to the **2a**–**5c** group, have less significant score values as kinase and/or protease inhibitors (score range 0.10–0.26) (see Appendix A for complete data).

Second, we determined the binding energies of BIZQs in their interaction sites with a set of known 3D structures of relevant proteins overexpressed in several cancer types as GC and BC [15,16,17,18,19,20,21,22], to identify the potential therapeutic targets for the BIZQs and their associated inhibitory interactions. Table 1 shows the predicted values of ΔG*_bin_* for their complexes with twelve selected proteins.

Table 1 shows that most of the BIZQs bind more strongly to COX-2 (3LN1), with ΔG*_bin_* values ranging from −10.4 to −8.7 kcal/mol (average −9.68), than to MAPK-1 (2OJG), with values ranging from −9.7 to −8.7 kcal/mol (average -9.14), and TPK-JAK (4EHZ), with values ranging from −10.4 to −8.0 kcal/mol (average −9.12). Some compounds, such as BIZQs **2a** and **3a**, and BIZQ **6k,** also showed their best values for VEGRF-2 (3VHE) and ER-α (3ERT), respectively. However, it should be noted that the best ΔG_bin_ value of −10.9 kcal/mol was found for the interaction of BIZQ **3a** with VEGRF-2.

As stated above, ∆G*_bin_* values between these compounds and COX-2 are better than those with other proteins overexpressed in GC and BC cell lines. Taking into account the ∆G*_bin_* average values (*Pavge.*), COX-2, MAPK-1 kinase, and TPK-JAK might be most prominent target proteins for the BIZQ derivatives than the rest of proteins shown in the Table 1. According to structural nature of the compounds tested, the best ΔG*_bin_* values observed for BIZQ derivatives of Series I with most of the evaluated proteins could be attributed to the absence of substituents at position N^1^ in the benzoindazole moiety, allowing H-bond formation with any oxygen or nitrogen atom present in the target proteins.

Considering the interaction of the BIZQs with all the proteins and their average ΔG*_bin_* values (*C avge.*), five BIZQs were the most relevant: (a) Those conjugated **6k** (−9.42 kcal/mol), **6c** (−9.33) and **6g** (−9.24), and (b) those simpler ones, **3a** (−9.11) and **2a** (−9.03) (Table 1). If we take into account only the three proteins with the best ΔG*_bin_* values, COX-2, MAPK-1, and TPK-JAK, their averages (*3 avge.*), correspond to **6k** (−10.03), **6c** (−10.03), **6g** (−9.93), **2a** (−9.78), **3a** (−9.67), and **2c** (−9.50), respectively (Table 1). Interestingly, in the BIZQs showing ΔG*_bin_* values ≤ −10 kcal/mol for COX-2, aromatic or unsaturated groups, as phenyl or prenyl, in the side chain are apparently important for the interaction of the molecules with COX-2.

Based on our results and the knowledge of the role of COX-2 in the inflammatory processes related to the development of GC [15,24], it is necessary to address further studies on the interaction of the BIZQs with COX-2. Besides their established antineoplastic cytotoxicity, the BIZQs could be used either as preventive or antimetastatic agents against GC, BC, and other inflammation-induced cancers. It should be noted that BIZQ **3a** showed a good ΔG*_bin_* value for VEGFR, a protein with an essential role during angiogenesis and carcinogenesis via the angiogenesis pathway [18]. A major regulator of angiogenesis is vascular endothelial growth factor (VEGF) and its associated receptor VEGFR-2, whose activation has been identified in several cancer processes [18,29]. Therapeutic agents targeting VEGF and VEGFR-2 have become a cornerstone of gastric and breast cancer, inhibiting cancer progression, and invasion into cell lines models [29,30,31,32]. Studies have shown that a high expression of COX-2 upregulates VEGFR expression, and that combination of selective COX-2 inhibitors (COXIBs) with VEGF angiogenesis pathway blockers could lead the control of metastasis in patients with colon cancer, breast cancer, and other tumors that overexpress COX-2 [33]. However, our results show that COX-2 presented better correlations with most of these BIZQs than VEGFR-2, which would suggest that COX-2 could be a promising therapeutic target for these compounds.

### 2.3. Binding Site and Docking of BIZQs in COX-2

In this work, a known docking screening protocol between the BIZQs and potential target proteins correlated with cancer was used, particularly with COX-2. To get further insights into the potential interaction, COX-2 binding sites were characterized in terms of their hydrogen bond networks, other binding interactions, and chemical moieties positions.

As indicated in Table 1, compounds of the three series showed similar ΔG*_bin_* values for COX-2, ranging from −9.4 (**5a**) to −10.4 (**2a** and **6c**) kcal/mol for Series I; from −8.7 (**4b**) to −10.1 (**6g**) kcal/mol for Series II; and from −9.3 (**4c**) to −10.2 (**6k**) kcal/mol for Series III. From a general point of view, these similar results suggested that COX-2-BIZQ interaction would not depend only on the substituents in the N^1^ position, which allow defining the different Series I–III or on the absence or presence of conjugated amino acids in the side chain enabling to differentiate into the simple ***subseries a*** (**2a**–**5c**) and the conjugate ***subseries b*** (**6a**–**6m**) of BIZQs. These findings would be related to the core structure of the BIZQs that would allow all of them to interact with the same binding pocket of COX-2.

To confirm such hypothesis, we performed the virtual BIZQs - COX-2 docking studies with the results shown in Table 2, where it can be observed that practically all the compounds lie in the same COX-2 binding cavity, and share a set of common amino acids in three main domains defined by the sequences Asn19-Phe49, Asp111-Cys145, and Gln447-Arg455. These domains include several amino acids involved in H-bonds interactions as Asn24, Asn28, Arg29, Cys32, and Ser34 within the first domain, Asp111, Tyr116, His119, Gly121, and Ala142 in the second, and Gln447 and Glu451 in the third domain, among others.

The epoxy derivatives **3a** and **3c** are the exceptions, because **3a** interacts with COX-2 in the domains His193-His200 and Asn368-His374, while **3c** interacts in the domains Asp333-Tyr341 and Ile550-Asn567, while the epoxide derivative **3b** interacts with COX-2 in the domains common to the rest of the BIZQs (Figure 2). These differences could be considered surprising for three compounds having the same side chain and suggest that the substituent at position N^1^, and particularly, the primary –OH group of the 2-hydroxyethyl fragment at such position plays an important role in the ligand-target interaction, because this group in the BIZQs **3b** and **4b** interact with the amide-carbonyl of His119 and with the amide-NH of Gly121 by H-bond contacts. Another interesting observation is that the interaction modes of **3b** and **4b** into the binding cavity of COX-2 are identical (Table 1, Figure 2), which confirm the isosteric character of the oxirane ring of **3b** and the carbonyl group of the aldehyde **4b** that interact with Gln447 by H-bonding.

Some illustrative examples of BIZQs-COX-2 docking are also presented for comparison purposes in Figure 3 and Figure 4. Figure 3 depicts the potential binding site of simple BIZQs and the poses of docked **2a**, **4a**, and **5a** into COX-2. The interactions of these compounds with the COX-2 binding pocket are governed by hydrogen bonds associated with the indazole-N-H, which interacts with the carbonyl oxygen of Asn24 present in the pocket of COX-2 at a distance of 2.00 Å, while only the quinone carbonyl at position C-9 of **2a** interacts with the NH_2_ group of Gln447 at 2.24 Å., That is due to its different spatial arrangement into the binding cavity with respect to **4a** and **5a**. The complexes are stabilized by hydrophobic interactions with different amino acids at the binding site of COX-2. For example, the **2a**- COX-2 complex would be stabilized by interactions with Cys21, Asn28, Arg29, Gly30, Glu31, Cys32, Tyr116, Leu138, Pro139, Glu451, Lys454, and Arg455 residues (Table 1). In addition, the interactions between COX-2 and BIZQs are also reinforced by Van der Waals forces, where others amino acids of the COX-2 binding pocket would take part in the correct orientation of the BIZQs into the pocket. 2D maps of some other BIZQs - COX-2 interactions can be seen in Appendix A.

Figure 4A shows the simultaneous docking of the simple BIZQ **2a** and the Phe conjugated BIZQ **6c**, and Figure 4B shows the independent and simultaneous docking of the three Phe conjugated BIZQs **6c**, **6g**, and **6k** into COX-2. The displacement of ligands in the pocket site and the different orientation of the indazole fragment of **2a** and **6c** (Figure 4A) are in agreement with the respective absence (**2a**) and presence (**6c**) of Phe in the side chain.

The docking differences observed in Figure 4B for the complexes of the three conjugated BIZQs **6c**, **6g**, and **6k**, which contain the same amino acid (Phe) at the side chain, are more striking. As seen, the indazole fragments of BIZQs **6c** and **6k** are respectively oriented up and down towards the inner part of the pocket, whereas with BIZQ **6g**, the whole benzoindazole moiety stays on the outside of the pocket. This change would be the consequence of the H-bonding observed between the hydroxyl proton of the hydroxyethyl substituent at N-1 of BIZQ **6g** and the Asp143 and Val141 residues of COX-2 (see 2D maps in Appendix A). Interestingly, it should be noted that despite the differences observed in docking geometries, the respective binding energies for the three interaction complexes, −10.4 (**6c**), −10.1 (**6g**), and −10.2 (**6k**) kcal/mol (Table 1), do not differ substantially between them.

### 2.4. In Vitro Cytotoxicity Results and their Correlation with Physicochemical and Pharmaco-Toxicological Parameters of BIZQs

Several of the synthesized BIZQs showed significant in vitro antiproliferative activity against KATO-III and MCF-7 cancer cells [14]. To establish any correlation between previous experimental results of in vitro antineoplastic cytotoxicity presented by BIZQs and the values obtained through docking and virtual screening, we carried out the evaluation of some physicochemical parameters, and the pharmacological and toxicological properties prediction for these derivatives. Table 3 shows the results expressed as pIC_50_ values of the in vitro anti proliferative evaluation of BIZQs against human KATO-III, GC, and MCF-7 BC cells using the MTS assay. The IC_50_ values in both types of cells are similar, with no statistically significant differences in cytotoxic potency.

Observations become apparent from the different compound/data arrangements shown in Table 3. According to this, the most potent group of the simplest BIZQs of the ***subseries a***, corresponds to those **2a**–**c** derivatives, which have the unsaturated prenyl group in the side chain. It should also be noted that these compounds are among those displaying the highest cytotoxic potency (IC_50_) and the best ΔG*_bin_* values and also show good drug-like scores. In parallel, when we consider the ***subseries b*** derivatives of the group of the BIZQs conjugated with Phe, with **6g** and **6k** as representative, showed the best ΔG*_bin_* and IC_50_ values, and the best drug-like scores. As can also be seen in Figure 1, compounds **2a**–**c**, **6g** and **6k**, have either an olefinic or a phenyl group at the side chain. Thus, the higher efficacy of these compounds could be attributed, in part, to the π-interactions of such groups with some amino acids of the target protein. For example, the olefin of **2a** has the estimated distances of 4,70 Å and 4,04 Å to the α-carbons of Asn28 and Arg455, respectively, while the phenyl group of **6k** has the estimated distance of 3.90Å to the α-carbon of Gly121 and of 3.93 Å to the phenyl group of Tyr122.

Interestingly, according to the Osiris-DataWarrior prediction, all the compounds of the ***subseries b*** (**6a**–**m**), and the half of those belonging to ***subseries a*** (**2a**–**c** and **5a**–**c**) are devoid of risks such as mutagenic, tumorigenic, reproductive-risk, and irritating effects. However, those simple BIZQs **3a**–**c** and **4a**–**c**, with oxirane and aldehyde functions, respectively, are predicted as potential promoters of such effects (Table 3), most probably due to their electrophilic nature and potential reactivity. Therefore, based on these prediction data, it would be necessary to confirm the toxic effects of the BIZQs through the corresponding experimental assays.

On the other hand, Table 3 shows that the pKa values cannot be used to deduce any structure/parameter-activity relationship, nor with the experimental in vitro results, nor with the calculated ∆G*_bin_* values, which would be a consequence of the structural variation within the series and subseries, of the reduced number of compounds within each subseries, and in many cases, due to the same magnitude of those pKa values predicted for similarly functionalized BIZQs.

Ibuprofen, non-selective COX inhibitor and traditional non-steroidal anti-inflammatory drug (NSAID), and Celecoxib, selective COX-2 inhibitor, and NSAID, were used as reference compounds [35]. ΔG*_bin_* values of Ibuprofen and Celecoxib for COX-2 were −7.7 and −8.8 Kcal/mol, respectively.

As illustrated in Figure 5, the predicted ∆G*_bin_* values for COX-2 are positively correlated with the ln[IC_50_] values in both KATO-III and MCF-7 cell lines. The correlation coefficients R of the Series I, II, and III for KATO-III are 0.72, 0.41, and 0.90, respectively and for MCF-7 the values are 0.79, 0.55, and 0.87, respectively. The observed results indicate that those BIZQs belonging to Series II, which contain a hydroxyethyl group at the N^1^ position of the indazole fragment, show the lowest correlation found between ln[IC_50_] and ΔG*_bin_* values for both KATO III and MCF-7 cancer cells. Particularly, in Series II, **5b** and **6e** compounds showed the best ΔG*_bin_*, but high IC_50_ values; therefore, the low cytotoxicity of these compounds could not be associated with their affinity for COX-2. The ∆G*_bin_* values for compounds of Series II and III correlated better with their respective ln[IC_50_] values obtained in MCF-7 than in KATO-III cell lines. Even though COX-2 is underexpressed in KATO-III cells, it is overexpressed in other gastric cancer cell lines as MKN-45. Accordingly, the IC_50_ values for these compounds could be related with a better interaction with other proteins, such as VEGRF-2 or ERs, that are also overexpressed in GC [19,21].

When we compare correlation ∆G*_bin_* vs. ln[IC_50_] between the simpler BIZQs (***subseries a***) and those with a conjugated amino acid residue in the side chain (***subseries b***), for KATO-III are 0.62 and 0.82 respectively, while for MCF-7 are 0.62 and 0.84, respectively. The R values observed for the two subseries would indicate that those BIZQs conjugated with C-protected amino acids show better correlations for both cell lines than the simpler BIZQs. Interestingly, the compounds with the best IC_50_ values (**2a**–**c**, **6g** and **6k**) for KATO-III and MCF-7 cell lines, have also better ∆G*_bin_* values than the other members of their corresponding subseries. It is worth noting that simple and the conjugated BIZQs show better ΔG*_bin_* values for COX-2 than those of NSAIDs with inhibitory activity on COX-2 (see footnote of Table 3). As mentioned above, these compounds share a double bond or an aromatic system in the side chain which could increase the complex stability through π-interactions with amino acids of the binding pocket. The double bond in the side chain of **2a** would interact with Asn28 and Arg455 while the aromatic π-electrons of Phe in **6k** would do it with Gly121 and Tyr122 (see Appendix A).

Furthermore, pIC_50_ values are also positively correlated with the predicted cLogP values (Figure 6). The observed R correlation of these parameters for Series I, II, and III are 0.69, 0.67, and 0.48, respectively, and for MCF-7 cell are 0.67, 0.78, and 0.47, respectively. When we correlate pIC_50_ against cLogP for both, KATO-III and MCF-7 cells, BIZQs of Series II show the best values of R, contrary to the results observed in the correlation of ΔG*_bin_* vs. pIC_50_. The R values for the correlation of pIC_50_ vs. cLogP of the ***subseries a*** and ***b***, for KATO-III are 0.77 and 0.72, respectively, and for MCF-7 are 0.80 and 0.74, respectively. If we consider the correlation within each subseries, we find that the R values are better than those between the series.

Interestingly, the low pIC_50_ value of compounds **5b**, **6a**, **6e**, and **6i**, both for KATO-III and MCF-7, could be explained by their reduced ability to cross the cell membrane, taking into account that their cLogP values are lower than those of the rest of BIZQs in their respective series. On the other hand, the simpler BIZQs **2a**–**c** and the conjugated BIZQs **6g** and **6k** show higher pIC_50_ values for both cell lines. These results could be associated to a better ability to cross the cell membrane, considering that their cLogP values are higher than those of the rest of BIZQs, and to their better binding affinity for COX-2.

### 2.5. In Silico ADME Studies

Prediction values for some pharmacokinetic parameters of the BIZQs derivatives related to oral absorption, Caco-2 cell permeability, blood-brain barrier permeability, and binding to human serum albumin, among others, are summarized in Appendix A. ADME descriptor values show that the percentage of predicted oral absorption de BIZQs varies from 30% to 93%, which indicates a poor to good oral bioavailability. Particularly, BIZQs **2a** and **3a** from Series I, **2b**, and **3b** from Series II, and **2c** and **3c** from Series III, have values higher than 80%. Accordingly, most BIZQ derivatives display good to excellent predicted values of Caco-2 cell permeability, except compounds **5a**, **6a**, and **6d** from Series I, compounds **5b**, **6e**, and **6h** from Series II, and compounds **4c**, **6i**, and **6m** from Series III. Besides, most BIZQ derivatives, except those conjugated compounds **6d**, **6h**, **6i**, **6k**, and **6m** would not cross the blood-brain barrier thus displaying a reduced risk of developing central nervous system (CNS) side effects. Also, all tested compounds were found within the range of interaction with human serum albumin, making possible their transport by plasma proteins to the target site. Globally, the BIZQs, with the only exception of **6k**, are assumed to have enough to excellent solubility in water, with logS values between −6.12 and −2.71. Almost all the BIZQs accomplish the Jorgensen’s rule of three, with the exceptions of **6k** and **6m**, which display 3 violations, though always within the permitted limits (see Appendix A).

The drug-like properties of the BIZQs are summarized in Appendix A. Practically, all the BIZQs accomplish the Lipinski’s rule of five and its Weber extension [36], with the exceptions of **6k** and **6m** (MW > 500), and **6h** and **6m** (rotatable bonds > 10), though also always within the permitted violations (see Appendix A). All these results would indicate that, from the pharmacokinetic point of view, most BIZQs could serve as good candidates for preclinical efficacy and toxicity assays.

## 3. Materials and Methods

### 3.1. Chemistry

The chemical procedures applied to obtain those here studied 1*H*-benzo[*f*]indazole-4,9-dione (BIZQ) derivatives **2a**–**c** to **5a**–**c** and **6a**–**m** was described in our previous article [11,14], while the route of synthesis for simple and conjugated BIZQs can be found in Appendix A. Briefly, the twenty-four benzoindazole derivatives, were synthesized by a direct cyclization reaction of 2-acetyl-6-(4-methyl-3-pentenyl)-1,4-naphthoquinone with hydrazines, followed by subsequent chemical modifications of the (4-methyl-3-pentenyl) chain through epoxidation, degradative oxidation, further oxidation, and *N*-acyl condensation reactions with protected amino acids, as previously described [14].

### 3.2. Computational Details

#### 3.2.1. Ligand Preparation

The 3D structure of each compound was built using Gaussview and geometrically optimized by the PM3 semiempirical methods using the Gaussian03 package [37]. These structures were visually checked to correct some structural errors. LogD and pKa values were obtained using ACD/Labs software [34].

#### 3.2.2. In Silico ADME Prediction

Pharmacokinetics parameters were calculated using QikProp (QP) v4.3 of the Schrödinger Suite [38]. Based on Lipinski’s rule of five and its extensions [34], we calculated molecular weight (mol, MW), logarithm of partition coefficient (QPlogPo/w), number of hydrogen bond acceptors (accptHB), number of hydrogen bonds donors (donorHB), number of rotatable bonds (#rotor), Van der Waals surface area of polar nitrogen and oxygen atoms (PSA). The pharmacokinetic profile was generated by the Jorgensen method and predicted for apparent Caco-2 cell permeability, brain/blood partition coefficient, binding to human serum albumin, apparent MDCK cell permeability, skin permeability, and qualitative human oral absorption [39].

#### 3.2.3. Macromolecules Selection and Retrieve

The crystal structure of 12 selected proteins (see Table 1), including enzymes, growth factor receptors, and transcription regulators, were retrieved from the Protein Data Bank [40]. They are overexpressed in some malignancies, including breast and gastric carcinomas, as described in the literature [15,16,17,18,19,20,21,22].

#### 3.2.4. Molecular Docking of Ligand-Protein Interaction

We resorted to virtual screening using Autodock Vina, a target-specific scoring method useful for virtual screening [41]. The three series of 1*H*-benzo[*f*]indazole-4,9-diones were docked into a set of proteins to identify the target protein potentially inhibited by these compounds. Both ligands and proteins were prepared using AutoDock Tools version 1.5.6 (ADT) according to the AutoDock Vina High Throughput screening standard method [41]. Gasteiger partial charges were assigned to the atoms of ligands. The AutoTors option was used to define the rotatable bonds in the ligands. The visual inspection of the results was performed using the Molecular Graphics Laboratory (MGL) Tools package. We selected a grid volume enough to cover each receptor. Finally, graphical analysis of molecular docking studies was performed using Visual Molecular Dynamics (VMD), version 1.9.2 [42]. The amino acids responsible for hydrogen bonding and hydrophobic interactions with the compounds were identified using LigPlot+ program as described by Wallace et al. [43]. Furthermore, molecular docking of reference compounds, ibuprofen and celecoxib, with COX-2 were done using the same procedure for the twenty-four benzoindazole derivatives.

### 3.3. Biological Activity

The cytotoxic effects of the 1*H*-Benzo[*f*]indazole-4,9-dione derivatives were analyzed by in vitro assay on KATO-III human gastric cancer cells and MCF-7 human breast adenocarcinoma cells, obtained from American Type Culture as described. Briefly, the efficacy of antitumor activity of each BIZQs was determined using MTS (colorimetric test), and the half maximal inhibitory concentration (IC_50_) was obtained from dose-response curves in both KATO-III and MCF-7 as previously described by Molinari et al. (2015) [14].

### 3.4. Statistical Analysis

In silico data are expressed as the means, while in vitro data are expressed as the means ± SEM for three independent experiments. The IC_50_(μM) obtained on KATO-III and MCF-7 cell lines was transformed into pIC_50_ (−log IC_50_). The degree of the linear relationship between two variables was measured using Pearson’s correlation coefficient (R). A value of *p* < 0.05 was taken as significant.

## 4. Conclusions

In conclusion, our work provides a rational molecular basis for identifying cancer-related proteins that could potentially be inhibited by the BIZQ (1*H*-benzo[*f*]indazol-4,9-dione) family of compounds. For this purpose, we evaluated potential protein targets for these compounds on KATO-III gastric carcinoma and MCF-7 breast carcinoma cell lines. The most prominent BIZQs were **2a**–**c**, **6k**, and **6g**, because they showed the best values of IC_50_ and ∆G*_bin_*. In this context, and as the most relevant fact, our results indicated that within those considered proteins, the inflammation-related COX-2 was the best target for the studied BIZQs, followed by the proliferation-related kinases, MAPK-1 and TPK-JAK, and the angiogenesis-related receptor VEGFR-2. Due to these significant discoveries, and to the abundant reports on the roles of COX-2 in the implantation, evolution and dissemination of inflammation-related cancer, additional studies must be conducted to ascertain the potential usefulness of BIZQs as preventive anticancer and antimetastatic agents. Consequently, it will be firstly necessary to validate and confirm experimentally the predictions and theoretical results found for the BIZQs and particularly for those being more potent or with better ∆G*_bin_* values of interaction with COX-2, and with the above mentioned proteins. Then, aiming to define the best candidates for pre-clinical evaluations and further development, a larger family of simple and conjugated BIZQs, including others with different aromatic amino acids, will be designed, synthesized and evaluated for structure optimization. The continuation of this research project is also positively supported by the results of the wide in silico evaluation and characterization of BIZQs reported here. Thus, the predictions on intrinsic bioactivity, drug-likeness scores, low toxicity risks, global good physicochemical, and pharmacokinetic parameters, which favorably correlated with previous in vitro anti-proliferative results, also point towards new and promising antineoplastic drugs candidates.

## Figures and Tables

**Figure 1 molecules-24-02261-f001:**
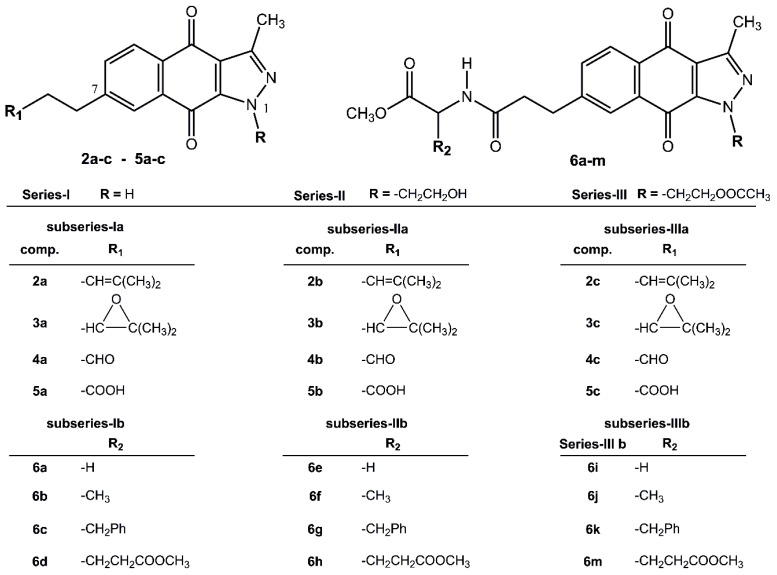
Structures of Series I, II, and III of 1*H*-benzo[*f*]indazole-4,9-quinones (BIZQs).

**Figure 2 molecules-24-02261-f002:**
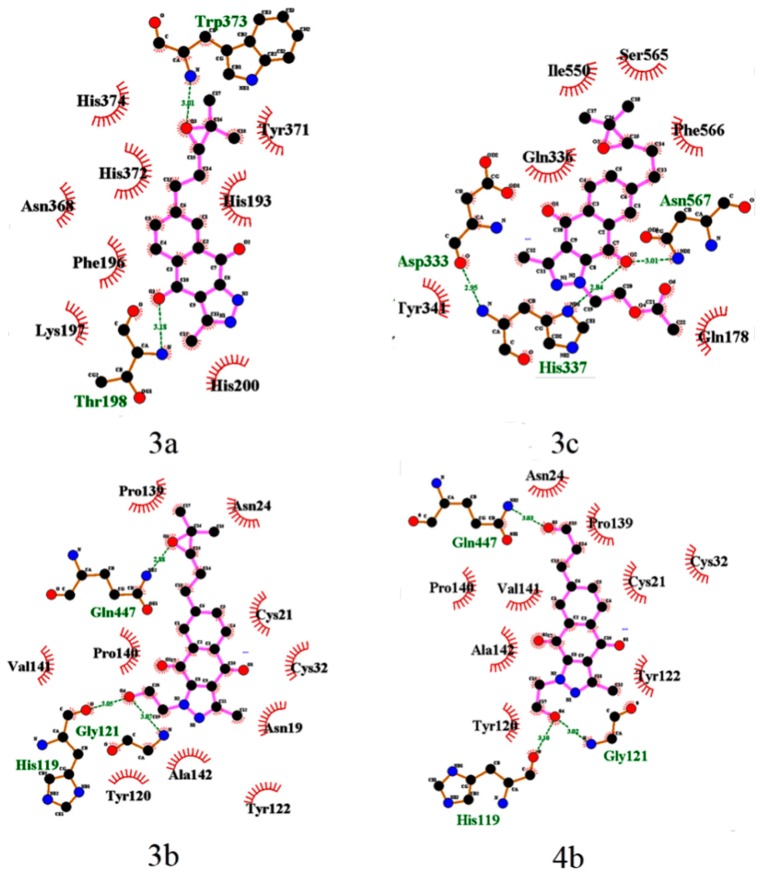
LigPlot+ 2D-maps of hydrogen-bond interaction patterns and hydrophobic contacts between some BIZQs and the main-chain or side-chain elements of COX-2 protein. The amino acids responsible for hydrogen bonds and hydrophobic interactions are represented by three-letter codes in green and black, respectively. Carbon, oxygen, and nitrogen atoms are represented by filled black, red, and blue circles, respectively.

**Figure 3 molecules-24-02261-f003:**
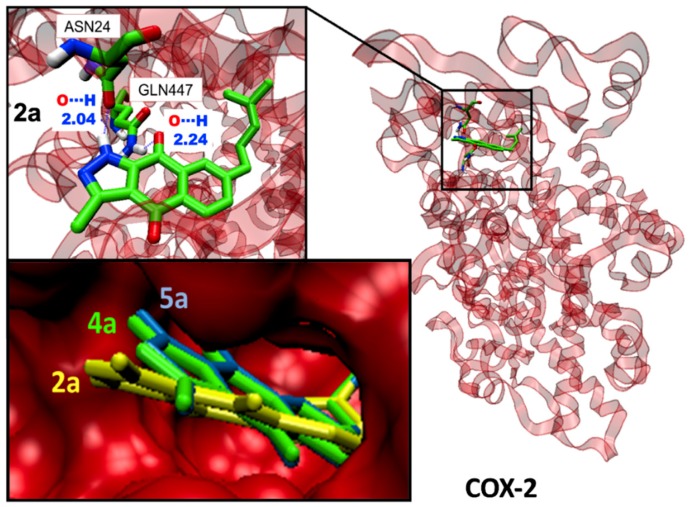
Visualization of the potential binding site and docking poses of BIZQs **2a**, **4a**, and **5a** into COX-2.

**Figure 4 molecules-24-02261-f004:**
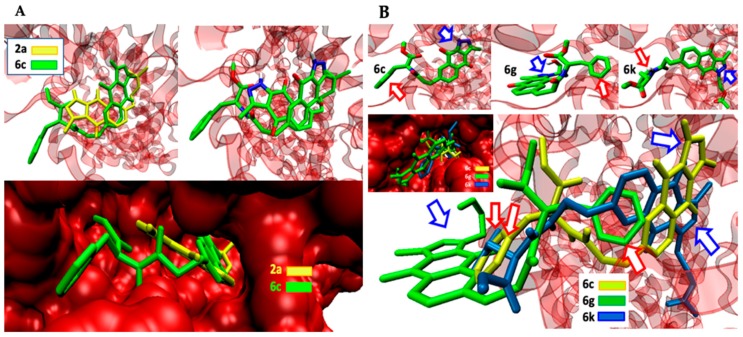
(**A**): Comparative arrangement of simple (**2a**) and conjugated (**6c**) BIZQs docked in COX-2. (**B**) Independent and simultaneous docking of BIZQs **6c**, **6g**, and **6k** conjugated with Phe. Blue and red arrows point to the pyrazole ring of each BIZQ and to the phenyl group of each Phe, respectively.

**Figure 5 molecules-24-02261-f005:**
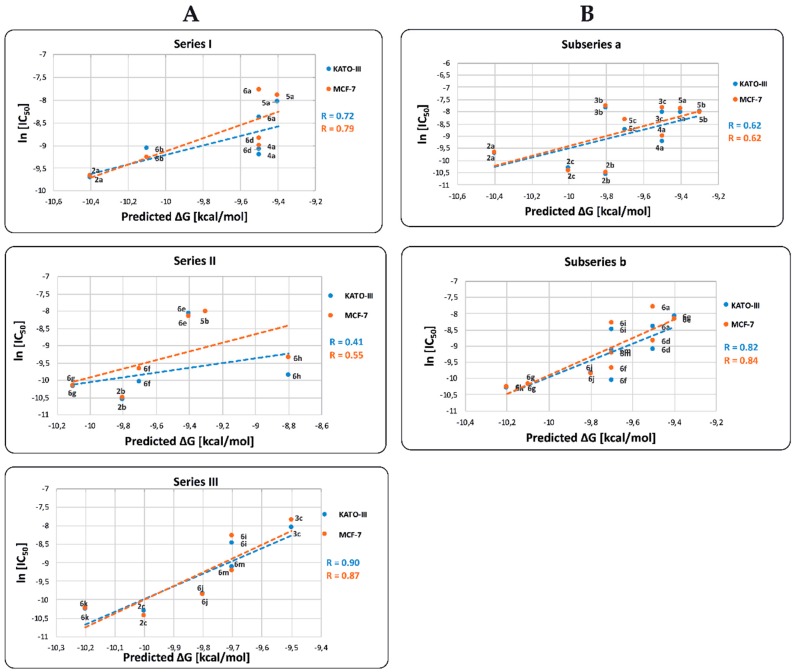
Relationship between ln[IC_50_] and binding free energy (ΔG*_bin_*) values of BIZQs for: (**A**) **Series I** (BIZQs that have no substituent at N), **II** (BIZQs that have a 2-hydroxyethyl group), and **III** (BIZQs that contain a 2-acetoxyethyl group); (**B**) ***subseries a*** (simpler BIZQs) and ***b*** (conjugated amino acids BIZQs) with COX-2 protein. Compounds **3a** and **6c** (Series I), **3b** and **4b** (Series II), **4c** and **5c** (Series III), **3a**, **4b** and **4c** (***subseries a***) and **6b**, **6c** and **6h** (***subseries b***) were excluded from the statistical analysis because they are outlier data points that significantly affect the correlation ln[IC_50_] versus ∆G*_bin_*.

**Figure 6 molecules-24-02261-f006:**
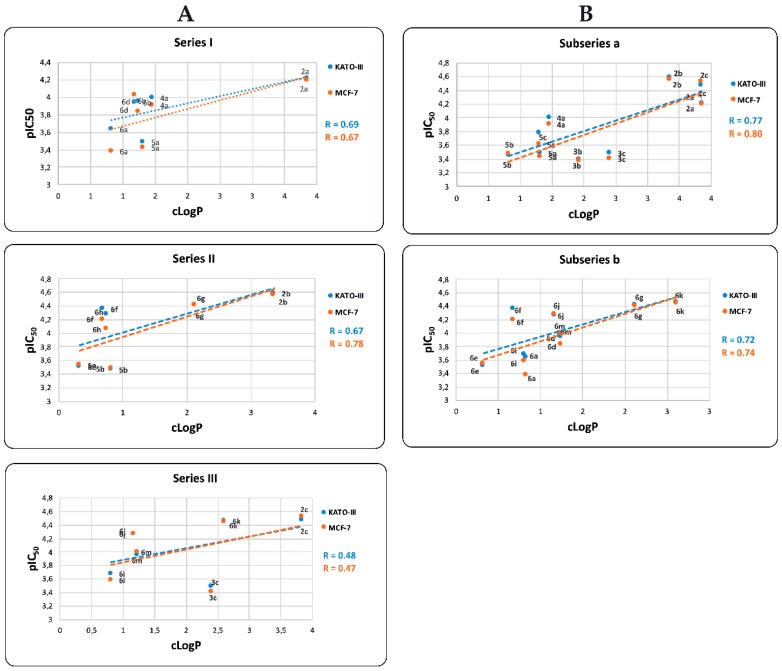
Relationship between pIC_50_ and cLogP values for BIZQ derivatives of (**A**) Series I, II, and III and (**B**) ***subseries a*** and ***b***. In this correlation analysis the same compounds as those considered in Figure 5 were taken into account.

**Table 1 molecules-24-02261-t001:** Predicted binding free energy values (ΔG*_bin_*, kcal/mol) of the BIZQs from Series I, II, and III for cancer-related proteins.

Compd.	PDB Entries												*C avge.*
	1DLS	1GJO	2OJG	2QTU	2ZZ0	3ERT	3LN1	3RXH	3VHE	4AN3	4EHZ	5GWK	
***Series Ia***													
**2a**	−7.2	−8.5	**−9.4**	**−9.8**	−8.4	−8.2	**−10.4**	**−9.1**	**−10.2**	**−9.3**	**−9.5**	−8.3	**−9.03**
**3a**	−8.9	−8.5	**−9.6**	−8.8	−8.6	−7.9	**−10.0**	**−9.1**	**−10.9**	**−9.3**	**−9.4**	−8.3	**−9.11**
**4a**	−7.7	−8.3	−8.8	−8.6	−8.5	−7.9	**−9.5**	−8.2	**−9.9**	−8.4	−8.8	−7.2	−8.48
**5a**	−8.0	−8.3	**−9.2**	**−9.0**	−8.8	−7.2	**−9.4**	−8.6	−8.4	−8.8	**−9.0**	−7.4	−8.51
***Series Ib***													
**6a**	−8.0	−8.4	**−9.1**	−8.2	**−9.3**	−7.5	**−9.5**	−8.2	−8.4	**−9.0**	**−9.1**	−6.9	−8.47
**6b**	**−9.0**	−8.5	−8.8	−8.4	**−9.4**	−7.8	**−10.1**	−8.2	−8.9	**−9.0**	**−9.2**	−7.5	−8.72
**6c**	**−10.0**	**−9.8**	**−9.4**	**−9.0**	**−9.5**	−8.5	**−10.4**	−8.4	−8.8	**−9.3**	**−10.3**	−8.5	**−9.33**
**6d**	−8.8	−8.6	−8.9	−7.8	−8.7	−8.0	**−9.5**	−7.9	−7.9	−8.8	**−9.0**	−6.7	−8.38
***Series IIa***													
**2b**	−7.6	−8.1	**−9.2**	−8.5	−8.2	−7.9	**−9.8**	−8.7	**−9.4**	−8.8	−8.7	−7.5	−8.53
**3b**	−8.6	−8.3	**−9.1**	−8.0	−8.5	−7.7	**−9.8**	−8.6	**−9.1**	**−9.0**	**−9.3**	−7.4	−8.62
**4b**	−7.6	−8.0	−8.9	−7.6	−7.9	−7.5	−8.7	−7.9	−7.9	−8.2	−8.0	−7.4	−7.97
**5b**	−8.0	−8.0	**−9.2**	−7.9	−8.3	−7.8	**−9.3**	−8.2	−8.3	−8.6	−8.9	−6.3	−8.23
***Series IIb***													
**6e**	−8.1	−8.1	**−9.2**	−7.9	−8.6	−7.7	**−9.4**	−8.3	−8.9	−8.9	−8.9	−7.2	−8.43
**6f**	−8.3	−8.3	**−9.3**	−7.8	−8.8	−7.8	**−9.7**	−8.3	−8.3	−8.6	−8.8	−7.4	−8.45
**6g**	**−9.0**	**−9.4**	**−9.7**	**−9.0**	**−9.8**	**−9.5**	**−10.1**	−8.7	−8.5	**−9.2**	**−10.0**	−8.0	**−9.24**
**6h**	−8.6	−8.4	−8.9	−7.8	−8.8	−8.1	−8.8	−8.0	−8.0	−8.0	−8.5	−7.4	−8.28
***Series IIIa***													
**2c**	−8.5	−8.4	**−9.3**	−8.5	−8.9	−7.9	**−10.0**	−8.3	**−9.6**	**−9.2**	**−9.2**	−7.5	−8.78
**3c**	−8.9	−8.4	**−9.5**	−8.2	**−9.0**	−8.0	**−9.5**	−8.5	**−9.2**	**−9.6**	**−9.4**	−7.7	−8.83
**4c**	−8.1	−8.8	−8.8	−7.2	−8.1	−7.2	**−9.3**	−7.9	−8.3	−8.4	−8.8	−6.7	−8.13
**5c**	−8.2	−8.4	**−9.1**	−7.8	−8.4	−7.6	**−9.7**	−8.3	−8.6	−8.8	−8.9	−6.9	−8.39
***Series IIIb***													
**6i**	−8.7	−8.4	**−9.0**	−7.9	**−9.1**	−8.1	**−9.7**	−8.2	**−9.2**	−8.7	**−9.0**	−7.3	−8.57
**6j**	−8.5	−8.3	−8.8	−8.1	**−9.5**	−8.2	**−9.8**	−8.3	**−9.2**	−8.9	−8.7	−7.6	−8.66
**6k**	**−9.5**	**−9.5**	**−9.5**	−8.9	**−9.6**	**−10.0**	**−10.2**	**−9.0**	−8.8	**−9.1**	**−10.4**	−8.5	**−9.42**
**6m**	**−9.0**	−8.6	−8.7	−7.9	**−9.2**	−8.4	**−9.7**	−7.8	−8.7	−8.7	−8.9	−6.8	−8.60
***P avge*.**	−8.50	−8.51	**−9.14**	−7.9	−8.83	−8.02	**−9.68**	−8.40	−8.55	−8.86	**−9.12**	−7.43	

PDB (Protein Data Bank) entry to each protein. 1DLS: Dihydrofolate reductase (DHFR); 1GJO: Fibroblast Growth Factor Receptor 2 (FGFR-2); 2OJG: Mitogen-activated protein kinase (MAPK-1); 2QTU: Estrogen receptor beta (NR3A2); 2ZZ0: Thioredoxin reductase 1 cytoplasmic (TXNRD1); 3ERT: Estrogen receptor α (NR3A1); 3LN1: Cyclooxygenase-2 (COX-2); 3RXH: Cationic trypsin (5GXP); 3VHE: Vascular endothelial growth factor receptor 2 (VEGRF-2); 4AN3: Mitogen-activated protein kinase (MAPKs); 4EHZ: Tyrosine-protein kinase (TPK-JAK); 5GWK: Topoisomerase II alpha (Topo IIA); ***C avge.***: Compound average, mean of the ΔG*_bin_* values of each compound with all 12 proteins. ***P avge.***: Protein average, mean of the ΔG*_bin_* values for the interactions of each protein with all the compounds. Absolute ΔG*_bin_* values ≥ 9 kcal/mol are highlighted in black, those ≥ **9.5** in blue and those ≥ **10** in red colors for easier affinity comparisons.

**Table 2 molecules-24-02261-t002:** Binding site contacts of BIZQs with partial amino acids sequences of COX-2.

BIZQ	H-Bonds and Hydrophobic Contacts in the Binding Site *
**Series Ia**	
**2a**	Cys21, **Asn24**, Asn28, Arg29, Gly30, Glu31, Cys32//Tyr116, Leu138, Pro139//Glu451, **Gln447**, Lys454, Arg455
**3a**	His193, Ph196, Lys197, **Thr198**, His200//Asn368, Tyr371, His372, **Trp373**, His374
**4a**	**Asn24**, Asn28, **Arg29**, Gly30, Cys32//Tyr116, Gly121, Leu138, Pro139//Lys454, Arg455
**5a**	**Asn24**, Gln27, **Asn28**, Arg29, Gly30, Glu31,Cys32//Tyr116, Leu138, Pro139//Gln447, Lys454, Glu451, Arg455
**Series Ib**	
**6a**	Asn19, Cys21, Cys22, Asn24, Cys26, Arg29, Gly30, Cys32//Tyr116, Leu138, Pro139, Pro140, Val141, **Ala142**//Gln447, Arg455
**6b**	Cys21, Asn24, Gln27, **Asn28**, **Arg29**, Gly30, Glu31, **Cys32//Gly121**, Leu138, Pro139//Gln447, Glu451, Lys454, Arg455
**6c**	Asn19, Cys21, Cys22, Asn24, Cys26, Arg29, Gly30, Glu31, Cys32//Tyr116, **Gly121**, Tyr122, Leu138, Pro139, Pro140, Val141, Ala142//Arg455
**6d**	**Asn19**, Cys21, Cys22, Asn24, Cys26, Arg29, Cys32//Tyr116, **Gly121**, Leu138, Pro139, Pro140, Val141, **Ala142**//Lys454, Arg455
**Series IIa**	
**2b**	Cys21, Cys22, Asn24, **Arg29**, Cys32, Gln44//**Asp111**, Tyr116, Gly121, Ala137, Leu138, Pro139, Pro140, Ala142, Cys145//Gln447, **Arg455**
**3b**	*Asn19*, Cys21, Asn24, Cys32, **His119**, Tyr120, **Gly121**,//Tyr122, Pro139, Pro140, Val141, Ala142//**Gln447**
**4b**	Cys21, Asn24, Cys32//**His119**, Tyr120, **Gln121**, Tyr122, Pro139, Pro140, Val141, Ala142//**Gln447**
**5b**	Cys21, **Asn24**, Cys26, Gly27, **Asn28**, Arg29, Gly30, Glu31, Cys32//Tyr116, Gly121, Leu138, Pro139//**Gln447**, Glu451, Lys454, Arg455
**Series IIb**	
**6e**	**Asn19**, Cys21, Asn24, Cys26, Arg29, Gly30, Glu31, Cys32//Tyr116, Gly121, Tyr122, Leu138, Pro139, Ala142//**Gln447**, **Glu451**
**6f**	**Asn19**, Cys21, Asn24, Cys26, Arg29, Gly30, Glu31, Cys32, Met33, **Ser34**//Tyr116, Gly121, Tyr122, Leu138, Pro139//**Gln447**, **Glu451**
**6g**	Asn19, Cys21, Cys22, Asn24, Cys32//Val118, His119, **Gly121**, Tyr122, Pro139, Pro140, **Val141**, **Ala142**, **Asp143**, Cys145
**6h**	**Asn19**, Cys21, Asn24, Cys26, Arg29, Gly30, Glu31, Cys32//Tyr116, His119, Gly121, Tyr122, Leu138, Pro139, Pro140, Ala142//**Gln447**, **Glu451**
**Series IIIa**	
**2c**	Cys21, Cys22, Asn24, Asn28, Arg29, Gly30, Glu31, Cys32//Tyr116, Gly121, Leu138, Pro139, Val141, Ala142//Gln447, Glu451, Lys454, Arg455
**3c**	Gln178//**Asp333**, Gln336, **His337**, Tyr341//Ile550, Ser565, Phe566, **Asn567**
**4c**	**Arg29**, Cys21, Asn24, Gln27, Asn28, Gly30, Glu31, Cys32//Tyr116, Gly121, Leu138, Pro139, Pro140, Val141, **Ala142**//Glu451, Lys454, Arg455
**5c**	**Arg29**, Cys21, Asn24, Gln27, Asn28, Gly30, Glu31//Tyr116, Gly121, Leu138, Pro139, Pro140, Val141, **Ala142**//Gln447, Glu451, Lys454, Arg455
**Series IIIb**	
**6i**	**Asn19**, Cys21, Gln27, Asn28, **Arg29**, Gly30, Glu31, Cys32//Tyr116, Gly121, Tyr122, Leu138, Pro139, Pro140, **Ala142**//Glu451, Lys454, Arg455
**6j**	**Asn28**, **Arg29**, Arg46, Thr47, Phe49//Tyr108, Leu138//Glu451, Lys454, Arg455
**6k**	**Asn19**, **Arg29**, Cys21, Asn24, Gln27, Asn28, Gly30, Glu31, Cys32//Tyr116, Gly121, Tyr122, Leu138, Pro139, Ala142, Asp143//Glu451, Lys454, Arg455
**6m**	**Asn19**, Cys21, Asn24, Gln27, Asn28, **Arg29**, Gly30, Glu31, Cys32, Met33, **Ser34**//Tyr116, **Gly121**, Tyr122, Leu138, Pro139, Ala142//Glu451, Lys454, Arg455

* The bolded names correspond to those amino acids involved in H-bonds with the corresponding BIZQ. Slash bars (//) used to define the partial amino acid sequences involved in contacts with every BIZQ.

**Table 3 molecules-24-02261-t003:** In vitro cytotoxicity results on KATO-III and MCF-7 cells, calculated binding energies, physicochemical parameters, drug-likeness scores, and potential toxicity risks for BIZQs.

BIZQ	In Vitro pIC_50_ ^a^		∆G_bin_ (kcal/mol)		Predicted Parameters		D-like	Toxicity Risks ^b^			
	KATO-III	MCF-7	COX-2	3avge.^c^	pKa1/pKa2 ^d,e^	cLogP ^f^		*M*	*T*	*R*	*I*
***Subseries a***											
**2a**	**4.23 ± 0.22**	**4.20 ± 0.17**	**−10.4**	**−9.78**	8.86/---	3.856	00.716	*n*	*n*	*n*	*n*
**2b**	**4.59 ± 0.18**	**4.56 ± 0.25**	**−9.8**	−9.23	14.94/---	3.354	22.137	*n*	*n*	*n*	*n*
**2c**	**4.48 ± 0.26**	**4.53 ± 0.14**	**−10.0**	−9.50	---/---	3.838	33.050	*n*	*n*	*n*	*n*
**3a**	3.50 ± 0.19	3.36 ± 0.18	**−10.0**	**−9.67**	8.86/---	2.425	00.351	***l***	***l***	***h***	*n*
**3b**	3.40 ± 0.16	3.38 ± 0.13	**−9.8**	−9.40	14.94/---	1.923	11.691	***l***	***l***	***h***	*n*
**3c**	3.49 ± 0.17	3.41 ± 0.10	−9.5	−9.47	---/---	2.408	22.564	***l***	***l***	***h***	*n*
**4a**	**4.00 ± 0.13**	3.91 ± 0.21	−9.5	−9.03	8.86/---	1.454	00.087	***h***	*n*	*n*	***h***
**4b**	**4.20 ± 0.19**	**4.36 ± 0.16**	−8.7	−8.53	14.94/---	0.952	11.479	***h***	*n*	*n*	***h***
**4c**	**4.22 ±0.13**	**4.48 ± 0.21**	−9.3	−8.97	---/---	1.436	22.196	***h***	*n*	*n*	***h***
**5a**	3.49 ± 0.12	3.43 ± 0.13	−9.4 ^g^	−9.20	3.17/8.86	1.317	22.104	*n*	*n*	*n*	*n*
**5b**	3.47 ± 0.10	3.48 ± 0.11	−9.3 ^g^	−9.13	3.15/14.94	0.815	33.465	*n*	*n*	*n*	*n*
**5c**	3.79 ± 0.15	3.61 ± 0.18	**−9.7 ^g^**	−9.23	3.17	1.299	44.378	*n*	*n*	*n*	*n*
***Subseries b***											
**6a**	3.64 ± 0.19	3.38 ± 0.20	−9.5	−9.23	8.86/11.52	0.828	**00.103**	*n*	*n*	*n*	*n*
**6e**	3.51 ± 0.16	3.54 ± 0.12	−9.4	−9.17	11.31/14.94	0.326	**11.123**	*n*	*n*	*n*	*n*
**6i**	3.68 ± 0.15	3.59 ± 0.15	**−9.7**	−9.23	11.32/---	0.811	**22.001**	*n*	*n*	*n*	*n*
**6b**	3.94 ± 0.13	**4.03 ± 0.14**	**−10.1**	−9.37	8.86/11.41	1.187	−1.153	*n*	*n*	*n*	*n*
**6f**	**4.36 ± 0.19**	**4.20 ± 0.24**	**−9.7**	−9.27	11.20/14.94	0.685	0.042	*n*	*n*	*n*	*n*
**6j**	**4.27 ± 0.26**	**4.28 ± 0.14**	**−9.8**	−9.10	11.21/---	1.170	0.867	*n*	*n*	*n*	*n*
**6c**	3.90 ± 0.21	3.81 ± 0.18	**−10.4**	**−10.03**	8.86/11.50	2.629	**2.272**	*n*	*n*	*n*	*n*
**6g**	**4.42 ± 0.22**	**4.41 ± 0.22**	**−10.1**	**−9.93**	11.32/14.94	2.127	**3.478**	*n*	*n*	*n*	*n*
**6k**	**4.46 ± 0.14**	**4.45 ± 0.27**	**−10.2**	**−10.03**	11.33/---	2.612	**4.317**	*n*	*n*	*n*	*n*
**6d**	3.95 ± 0.21	3.84 ± 0.16	−9.5	−9.13	8.86/11.39	1.248	−3.119	*n*	*n*	*n*	*n*
**6h**	**4.28 ± 0.21**	**4.06 ± 0.12**	−8.8	−8.73	11.22/14.94	0.746	−1.914	*n*	*n*	*n*	*n*
**6m**	3.96 ± 0.23	4.00 ± 0.12	**−9.7**	−9.10	11.22/---	1.230	−1.060	*n*	*n*	*n*	*n*

a: IC_50_ (μM), Half-maximal inhibitory concentration; pIC_50_ = −log IC_50_ [14]. Values of pIC_50_, > 4 are highlighted for comparison. Absolute ΔG*_bin_* values > 9.5 kcal/mol and Drug-likeness scores (**D-like**) > 2, are also bolded. b: Predicted through DataWarrior algorithms [28], ***M***: Mutagenic, ***T***: Tumorigenic, ***R***: Reproductive effective, ***I***: Irritant; levels: None (*n*), low (*l*) and high (*h*). c: **3avge**. Mean of the three ΔG*_bin_* values corresponding to the interactions of every BIZQ with COX-2, MAPK-1 and TPK-JAK. d, e: ACDLabs [34]. f: pKa values of 8.86, 11.2–11.5, 3.15–3.17, and 14.94 are associated with the ionization of indazole-*N*-H, side-chain-amide-*N*-H, carboxyl-*O*-H and hydroxyl-*O*-H, respectively. g: values for non-ionized carboxylic acids.

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
