# Peer review of "Antiproliferative Benzoindazolequinones as Potential Cyclooxygenase-2 Inhibitors"

_molecules, 2019, doi:10.3390/molecules24122261_

Round 1

Reviewer 1 Report

The presented results are interesting.  The work is a continuation of the research described in J. Heterocycl. Chem. 2015 and  Mocelules 2015.  In the experimental part regarding the synthesis of these compounds, this should be strongly emphasized. Figure 1 needs to be corrected: at compounds 5a: it should be COOH to show the carboxyl group and at 6d: COOCH3 should be introduced to show the ester moiety.  In Table 1 (concerning biological tests) a reference compound should be given.  Under the table 3 there should be an explanation of the pIC50, which is in part 3.4. The publications 11 should be improved : J. Heterocycl. Chem. 2015, 52 , 620-622 !!  Please, check again carefully quoted publications (pages, volumes).

Author Response

We sincerely appreciate the valuable comments of reviewer 1 (in Black color) and respond (in RED color), accordingly to the questions below:

Point 1: The presented results are interesting.

Response 1: OK.   No response/correction needed

Point 2: The work is a continuation of the research described in J. Heterocycl. Chem. 2015 and Molecules 2015.      

Response 2: OK.   No response/correction needed

Point 3: In the experimental part regarding the synthesis of these compounds, this should be strongly emphasized

Response 3: As mentioned by the Reviewer, the full description of the synthesis and characterization of BIZQ was published in Molecules 2015, 20, 21924-21938. We think that it would not be advisable to include such information again in this manuscript, assuming also that the included structures would serve well enough to follow the studies and the results reported. However, according to the reviewer's suggestion, in order to facilitate the monitoring of the chemical aspects of the research, we have now added the schemes of the synthesis routes as supplementary material SS1 (Suppl. Scheme 1), including the corresponding mention in the text (3.Materials and Methods - 3.1.Chemistry section and in Supplementary Materials section).

Point 4: Figure 1 needs to be corrected: at compounds 5a: it should be COOH to show the carboxyl group and at 6d: COOCH3 should be introduced to show the ester moiety.

Response 4: Figure 1 was corrected, i.e. CO2H was replaced by COOH in the compounds 5a-c and CH2CH2CO2CH3 was replaced by CH2CH2COOCH3 in the compounds 6d, 6h and 6m.

Point 5: In Table 1 (concerning biological tests) a reference compound should be given.

Response 5: We agree with the reviewer’s suggestion. Ibuprofen, non-selective COX inhibitor and traditional non-steroidal anti-inflammatory drug (NSAID), and Celecoxib, selective COX-2 inhibitor and NSAID, were put as reference compounds in footnote of Table 3. ΔGbin values of ibuprofen and celecoxib for COX-2 were -7.7 and -8.8 Kcal/mol, respectively. The calculated ΔGbin were compared with those of BIZQs and discussed in Results & Discussion section

Point 6: Under the table 3 there should be an explanation of the pIC50, which is in part 3.4.

.

Response 6: Definition of pIC50 has been added under Table 3 and in Materials and Methods (points 3.3 and 3.4).

Point 7: The publications 11 should be improved: J. Heterocycl. Chem. 2015, 52, 620-622 !! Please, check again carefully quoted publications (pages, volumes).

Response 7: All references have been thoroughly revised and some of them adequately corrected.

Apart from all, the above-stated modifications, the typographical errors and the language correction suggested by the reviewers have been thoroughly corrected in the revised version.

Reviewer 2 Report

This work describes the analysis of the results obtained for the group of 24 benzoindazolequinone derivatives (BIZQs). Compounds were previously obtained, their syntheses described as well their evaluated antiproliferative in vitro activity using KATO-III and MCF-7 cell lines. In the present manuscript Authors performed in silico several experiments: virtual screening for potential antineoplastic targets- dihydrofolate  reductase, fibroblast growth factor receptor-2, mitogen-activated protein kinase, estrogen receptor beta, cyclooxygenase-2, cationic trypsin, vascular endothelial growth factor receptor-2,mitogen-activated protein kinase, tyrosine-protein kinase, topoisomerase II alpha. The structures of proteins were taken from PDB. There were not their crystal structures and docked ligands crystal structures available. Authors have modeled their structures and then calculated binding energies of BIZQs to their potential targets. As the best interaction with COX-2 was chosen. Interaction with particular amino acids was discussed. ADME properties and physicochemical properties of BIZQs were evaluated in silico. In conclusions Authors stated that on their results it was possible to define the best candidates for preclinical evaluations and further development. My comments:

Conclusions are overestimated. It is not proven on which target compounds are acting. If calculated the in silico binding it should be experimentally evaluated. All the performed experiments were in silico. If such analysis allowed to describe the features of compounds with expected activity (leading structures) then such structures should be obtained and evaluated. From this work it is not clear which structure elements and physicochemical properties should possess compounds with the expected activity. From the in silico and even  in vitro studies it is a very long way to preclinical evaluations

Author Response

We sincerely thank the reviewer 2 for reviewing our manuscript, their comments (in BLACK color). Our responses (in RED color) accordingly to the comments are listed below:

Comments author:

Point 1: This work describes the analysis of the results obtained for the group of 24 benzoindazolequinone derivatives (BIZQs). Compounds were previously obtained, their syntheses described as well their evaluated antiproliferative in vitro activity using KATO-III and MCF-7 cell lines. In the present manuscript Authors performed in silico several experiments: virtual screening for potential antineoplastic targets- dihydrofolate  reductase, fibroblast growth factor receptor-2, mitogen-activated protein kinase, estrogen receptor beta, cyclooxygenase-2, cationic trypsin, vascular endothelial growth factor receptor-2,mitogen-activated protein kinase, tyrosine-protein kinase, topoisomerase II alpha. The structures of proteins were taken from PDB Database. There were not their crystal structures and docked ligands crystal structures available. Authors have modeled their structures and then calculated binding energies of BIZQs to their potential targets. As the best interaction with COX-2 was chosen. Interactions with particular amino acids were discussed. ADME properties and physicochemical properties of BIZQs were evaluated in silico.

Response 1:   OK.   No response/correction needed

Point 2: In conclusions Authors stated that on their results it was possible to define the best candidates for preclinical evaluations and further development.

My comments: Conclusions are overestimated. It is not proven on which target compounds are acting. If calculated the in silico binding it should be experimentally evaluated. All the performed experiments were in silico. If such analysis allowed to describe the features of compounds with expected activity (leading structures) then such structures should be obtained and evaluated. From this work it is not clear which structure elements and physicochemical properties should possess compounds with the expected activity. From the in silico and even in vitro studies it is a very long way to preclinical evaluations.

Response 2: We have previously reported the design, synthesis and antiproliferative activity evaluation of a larger family of simple and conjugated-amino acids BIZQs. Our work provides a rational molecular basis for identifying potential molecular targets for these BIZQs. From docking studies with a set of cancer-related proteins, we identify that COX-2, as well as other relevant proteins, are the most suitable targets for BIZQs. Furthermore, we mentioned in the manuscript that it will be firstly necessary to validate and confirm experimentally the predictions and theoretical results found for the BIZQs and particularly for those being more potent or with better ∆Gbin values of interaction with COX-2. Furthermore, from docking and predicted physico chemical properties results, we indicated that the presence of quinone and heterocyclic systems, prenyl group, substituent at position N1 and phenyl groups in conjugated BIZQs are important structural elements in the interaction with target proteins and it would facility their arrival to target site. All these promising results are aiming to define the best candidates for preclinical evaluations.

According to your comments and suggestions, “Conclusion section” has been thoroughly reorganized in revised manuscript, aiming to achieve well-understanding.

Reviewer 3 Report

In this manuscript, the authors provide a rational molecular basis for identifying cancer-related proteins that could potentially be inhibited by the BIZQ (1H-benzo[f]indazol-4, 9-dione) family of compounds. The research work in this manuscript is very systematic and comprehensive, also very interesting. Thus, I would like to suggest this manuscript be accepted after minor revisions are made.

1.  That would be great if the authors can provide the synthetic routes to compounds 2a-c, 5a-c and 6a-m from corresponding starting materials in Chemdraw files. For most of the readers, especially synthetic chemists, organic chemists and medicinal chemists, it is very interesting and important to provide the synthetic routes to above mentioned compounds.

2.  In Figure 2, 3 and 4, detailed docking information should be provided. For example, how much ns is used when the complex of COX-2 with 6c, 6g and 6k were simulated in Figure 4, respectively. 10 ns? 20ns? or higher ns?

Author Response

Sincerely thanks to reviewer 3 for his valuable comments.  Our responses (in RED color) to all the author comments (in BLACK color) are listed below:

Reviewer Comments:

In this manuscript, the authors provide a rational molecular basis for identifying cancer-related proteins that could potentially be inhibited by the BIZQ (1H-benzo[f]indazol-4, 9-dione) family of compounds. The research work in this manuscript is very systematic and comprehensive, also very interesting. Thus, I would like to suggest this manuscript be accepted after minor revisions are made.

Point 1: That would be great if the authors can provide the synthetic routes to compounds 2a-c, 5a-c and 6a-m from corresponding starting materials in Chemdraw files. For most of the readers, especially synthetic chemists, organic chemists and medicinal chemists, it is very interesting and important to provide the synthetic routes to above mentioned compounds.

Response 1: The complete description of the synthesis and characterization of BIZQs was previously published in Molecules 2015, 20, 21924-21938 (Ref. 14). Also assuming that the included structures in Figure 1, would serve enough to follow the studies and results being reported. However, in order to facilitate the monitoring of the chemical aspects of the research, we have considered the Reviewer suggestion, and we have added the  syntheses schemes  in CHEMDRAW files as supplementary material SS1 (Suppl. Scheme 1). We have included a mention in the text (3.Materials and Methods- 3.1. Chemistry section and in Supplementary Materials section).

Point 2: In Figure 2, 3 and 4, detailed docking information should be provided. For example, how much ns is used when the complex of COX-2 with 6c, 6g and 6k were simulated in Figure 4, respectively. 10 ns? 20ns? or higher ns?

Response 2: We thank the reviewer for this suggestion. However, the docking procedure was adequately detailed in Materials and Methods. Furthermore, we decided not to do molecular dynamics simulations after docking because docking algorithm found expected ligands pose, as well as their molecular interactions, into binding site of the receptor and the ligands were docked to proteins with X-ray crystal structure taken from PDB database.

Apart from all the above-stated modifications, all the typographical errors and the language correction suggested by the reviewers have been thoroughly corrected in the revised version.

Round 2

Reviewer 2 Report

After revision I accept the manuscript